# An Early Warning System for Earthquake Prediction from Seismic Data Using Batch Normalized Graph Convolutional Neural Network with Attention Mechanism (BNGCNNATT)

**DOI:** 10.3390/s22176482

**Published:** 2022-08-28

**Authors:** Muhammad Atif Bilal, Yanju Ji, Yongzhi Wang, Muhammad Pervez Akhter, Muhammad Yaqub

**Affiliations:** 1College of Instrumentation & Electrical Engineering, Jilin University, Changchun 130061, China; 2College of Geoexploration Science & Technology, Jilin University, Changchun 130061, China; 3Institute of Integrated Information for Mineral Resources Prediction, Jilin University, Changchun 130026, China; 4Riphah College of Computing, Faisalabad Campus, Riphah International University, Faisalabad 38000, Pakistan; 5Faculty of Information Technology, Beijing University of Technology, Beijing 100021, China

**Keywords:** batch normalization, deep learning, attention layer, earthquake prediction, graph convolution network, the seismic network

## Abstract

Earthquakes threaten people, homes, and infrastructure. Early warning systems provide prior warning of oncoming significant shaking to decrease seismic risk by providing location, magnitude, and depth information of the event. Their usefulness depends on how soon a strong shake begins after the warning. In this article, the authors implement a deep learning model for predicting earthquakes. This model is based on a graph convolutional neural network with batch normalization and attention mechanism techniques that can successfully predict the depth and magnitude of an earthquake event at any number of seismic stations in any number of locations. After preprocessing the waveform data, CNN extracts the feature map. Attention mechanism is used to focus on important features. The batch normalization technique takes place in batches for stable and faster training of the model by adding an extra layer. GNN with extracted features and event location information predicts the event information accurately. We test the proposed model on two datasets from Japan and Alaska, which have different seismic dynamics. The proposed model achieves 2.8 and 4.0 RMSE values in Alaska and Japan for magnitude prediction, and 2.87 and 2.66 RMSE values for depth prediction. Low RMSE values show that the proposed model significantly outperforms the three baseline models on both datasets to provide an accurate estimation of the depth and magnitude of small, medium, and large-magnitude events.

## 1. Introduction

Several natural hazards like fire, tsunami, flood, earthquake, etc., are damaging humans, buildings, animals, and other infrastructures. These hazards can not be stopped but can be prevented using several techniques like remote sensign [1], LiDAR [2], and seismic stations [3]. One of the goals of earthquake hazard analysis is to limit the impact of earthquakes on society by minimizing damage to buildings and infrastructure. Predicting seismic events from continuous data and applying it in real-time to early warning systems or analyzing it offline to look for previously unpredicted earthquakes is an important task. Conventional algorithms can easily predict earthquakes of moderate or large magnitudes because they are so infrequent and scarce in seismic catalogs. A small magnitude earthquake may go unnoticed by expert analysts or automatic detection systems if, for example, the traces have a low signal-to-noise ratio (SNR) or overlapping events are recorded. Thus, small magnitude earthquakes may be missing from earthquake inventories. Early automatic earthquake prediction from raw waveform data collected by seismic station sensors has been a significant study subject in recent years for emergency response [4]. Earthquake early-warning (EEW) systems serve this function by generating instantaneous alerts to potential hazards in high-risk locations within seconds of an earthquake’s waves being detected. Computational technologies based on machine learning hold great promise for earthquake prediction.

Numerous automated tasks, such as text processing [5,6], image processing [7,8], and speech recognition [9,10] have shown that traditional machine learning and deep learning methods perform well. One of the main issues with the traditional machine learning model is their reliance on feature selection methods. However, numerous comparison studies reveal that there is no global feature selection method that is effective with all kinds of traditional machine learning models [11,12]. Unlike traditional machine learning models, deep learning models use hidden layers like convolutional, activation, pooling, batch normalization, and attention layers to automatically extract important and useful features for a range of tasks. Instead of depending on manually selected features, deep learning models automatically extract useful and significant features from the raw input data [13]. This feature allows deep learning models to utilize all of the information contained in an event’s waveforms. The aim of the EEW system is to issue an alert within seconds if eartquake waves are detected. To achieve this aim, deep learning models use their hidden layers to quickly process waveform data after receiving it from seismic stations and to predict earthquakes efficiently. Several recent studies show that deep learning models on raw seismic waveforms can effectively predict earthquake parameters like magnitude [14,15,16], location [16,17], and peak ground acceleration [14].

Hyper-parameter tuning is a difficulty with deep learning models. Some of the parameters of a model include the number of epochs, the number of convolutional filters, and the batch size. When it comes to parameter tuning, finding the optimal values for these parameters might take a lot of time and effort. To train deep learning models more reliably and quickly, batch normalization is employed [18,19]. Batch size refers to the number of examples given at a time to the model for processing in a single iteration. Small batch sizes improve training by requiring less memory and time [20]. Batch normalization is a process that makes a model faster and more stable by adding an extra layer to the model. This batch normalization layer performs standardizing and normalizing operations on the output of the previous layer. The normalizing process in batch normalization takes place in batches, not as a single input [7].

In convolutional neural networks (CNN), a feature map is obtained from a convolutional layer that has contextual information among the features but does not focus on the important feature. According to CNN, the attention process can be thought of as a way to improve quality by picking out the most beneficial parts of a wave [15,21]. We can adjust feature map weighting by the attention value after computing the attention value based on relevance or saliency in the feature map produced by the convolution layer [22]. Including batch normalization and attention layers in CNN helps to improve training and prediction accuracy [21].

In recent years, CNN [8,16], graph neural networks (GNN) [23], and their ensemble model graph convolutional neural networks (GCNN) [4,8] have shown promising results for earthquake prediction. GNNs are particularly designed to analyze data from networks. Nodes of a graph represent the stations of a network and the edges among nodes represent connections among stations [24]. The raw seismic waveform data is used by CNN to extract significant information. Convolutional layers in CNN extract contextual local features from the input waveform data while the pooling operation makes it possible to learn global features [21,25]. Accurate earthquake predictions are made possible by combining these features with the geographic information provided by the stations [14]. Faster and more stable model training can be achieved by implementing batch normalization in CNN [18]. CNN with an attention mechanism can be thought of as a way to improve characteristics by isolating the most useful parts of a signal [22,26]. We may obtain refined feature map weighting by the attention value after computing the attention value by the attention layer based on importance or saliency in the feature map generated from the convolution layer [22]. What to focus on and where to focus are the two major components of the attention process [21].

Deep learning algorithms have been used in the past to predict earthquakes. For seismic source characterization, Ref. [16] employed a GNN approach and appended latitude and longitude values to the feature maps extracted by CNN. For earthquake prediction, a graph partitioning approach using CNN has been proposed in Ref. [27]. These methods only employed conventional graph theory rather than the true GNN technique. Recently, Ref. [4] conducted a study that combined GNN and CNN to classify seismic events from multiple stations without using any geographical information or meta-data about the stations. However, the models that have been proposed publicly so far use fixed time windows of input waves and do not process data with different lengths. In a similar way, all models, with the exception of the model by Ref. [16], either process waveforms from a single seismic station or depend on a fixed set of seismic stations defined at training time. The model by Ref. [16] allows for the usage of a variable station set, however, it uses a simple pooling technique to aggregate measurements from various stations.

Therefore, in this study, we propose a GCNN model with batch normalization and attention mechanisms for earthquake prediction. The CNN component of the model uses a convolutional layer to extract valuable and important information from seismic raw waveform data gathered from multiple stations. The batch normalization layer, which comes before the activation layer, enhances learning and shortens training [18,19]. The convolutional layer’s information is used by the attention layer to enhance the feature map and identify the signal’s most valuable components. GNN analyses spatial data and meta-data about the base stations. In this way, the proposed model can accurately and efficiently predict earthquakes by processing seismic data from multiple station networks.

We made the following contributions to this study, in brief:We propose a graph convolutional neural network with batch normalization and attention mechanisms for the early prediction of earthquake magnitude and depth;For experiments, we use two datasets with different seismic dynamics; Alaska with sparse events distribution and Japan with dense events distribution;By carefully adjusting many hyper-parameters of our model, we have systematically analyzed how well it performs;We selected the GCNN [16], GCNN with batch normalization (BNGCNN), and GCNN with attention mechanism (GCNNAtt.) as baseline models to compare the results obtained from the proposed model.

The rest of this article has been organized as follows: in Section 2, we discuss related studies employing seismic data, convolutional networks, and graph networks. In Section 3, the architecture of the suggested model and the baseline models is discussed. A brief summary of the issues, the comparison, and the two seismic datasets are given in Section 4. The experimental results for the proposed model and the baseline models are presented in Section 5. The conclusion and the future work are included in Section 6.

## 2. Related Work

The concept of earthquake early warning has been known for more than a century, but the tools and procedures needed to implement it have just recently been created in the previous three decades. Early warning systems’ main goal is to alert users if shaking levels that could result in damage are about to happen [28]. Deep learning techniques have become popular in recent years as a useful tool for the quick assessment of earthquakes [14,29].

Traditional machine learning models like decision trees [30], naïve Bayes [30], and support vector machines [31] use different feature selection techniques like information gain and gain ratio to extract meaningful features from data for classification. The performance of these models is significantly impacted by these feature selection techniques [17,32]. Furthermore, employing feature selection techniques on unprocessed data is more labor-intensive and error-prone. In deep learning models, several hidden layers are used to automatically choose features to generate a feature map [33]. When these layers are used for dimensionality reduction, deep learning models can analyze complex and nonlinear inputs more successfully. Deep learning models like graph neural networks [4,34], and multilayer perceptrons [16] effectively capture spatial information, such as stations and their interactions.

On a range of tasks, including image processing [7,8], text classification [5,6], speech recognition [9,10], and others, when it comes to processing complex and large datasets efficiently and accurately, deep learning models have been shown to work better than traditional machine learning models. CNN, GNN, and RNN are the three primary architectures utilized in deep learning [8]. These models have primarily been inherited from the machine learning model multi-layer perceptron (MLP). MLP is a powerful fully-connected layered architecture with input, hidden, and output layers, but the depth of the model is restricted because of the significant amount of computational time required by the algorithm. To understand complex data and enhance performance, the advanced design of MLP has incorporated many layers of different types (convolutional, pooling, dropout, and softmax) [33].

CNN-based models have been used in some studies for data classification, processing seismic data, and extracting seismic features [23,35]. Jozinović et al. [35] used CNN to predict earthquakes and earthquake epicenter categorization from a single station waveform data. The proposed CNN model predicts earthquake sources over a wide range of distances and magnitudes with an accuracy of 87%, according to tests utilizing three-component waveform data collected from the USGS. Jozinović et al. [35] employed CNN to forecast the degree of ground shaking caused by an earthquake after 15–20 s had passed since the time the earthquake originated. According to Ref. [23], two CNN-based models were developed to assess the seismic reaction of the surface. In their study on earthquake prediction and source region estimation [3], the authors utilized a deep CNN model. Both the amplitude and the natural periods can be reasonably well predicted by the models that have been developed. These studies have all relied on waveform data collected from a single base station and do not take into account geographical information in their predictions.

In recent times, a hybrid deep learning model known as a graph convolutional neural network (GCNN) has gained popularity for its ability to accurately forecast earthquakes by exploiting seismic data. In Refs. [8,24], they used the GNN model to demonstrate how sensor position data, when combined with time-series data, may be utilized by graph-based networks. They used graph-based networks, which served as proof of this. The results of studies conducted on two seismic datasets that include earthquake waveforms point to promising outcomes. A deep CNN and GNN model, known as GCNN, was developed by Ref. [4] and Ref. [16] for the classification of earthquake events using a large number of stations. When combined with the spatial data provided by the stations, the features that the CNN layers extract from the waveform data help the GNN provide accurate earthquake predictions. Based on geophysical array data, a model using CNN and GNN to predict earthquakes was put out by Ref. [27]. This model is based on graph partitioning technique. Although they utilized data from numerous seismic stations, they completely disregarded the geographic information of these seismic stations.

Multiple stations in various locations can pick up on a variety of events. There is a connection between the observations that were gathered from the various stations, even if it is possible to generate diverse data due to the presence of multiple stations [4]. A great amount of consistent data can be generated from a single station’s measurements. As a result, earthquake prediction using data from numerous stations is a more difficult challenge for deep learning models to accomplish than earthquake prediction using data from a single station [24]. While CNN-based models are frequently utilized for processing single-station datasets [14,29], GCNN models are effective for the processing of multiple-station datasets [8,16]. In this study, our aim was to design a method for quick and reliable estimation of an earthquake’s magnitude directly from raw seismograms recorded on a single station.

One of the drawbacks of deep learning models is the difficulty of fine-tuning the parameters. Batch normalization is a technique that can be used to speed up and improve the accuracy of deep learning model training [18,19]. The goal of the batch normalization method during training is to normalize the layer output by utilizing the statistics of each mini-batch size. Recently, numerous studies have presented deep learning models and used batch normalization to enhance the model performance in areas such as the detection of brain tumors [7], the construction of gas-liquid interfaces [36], and the diagnosis of machine faults [7,37]. Unfortunately, batch normalization has not been used for automatic earthquake identification from seismic data provided by several stations.

Table 1 provides a synopsis of the research on the subject of earthquake prediction that was covered in this section. CNN and GCNN have spent the majority of their resources investigating the matter, but there are still a few questions that need to be answered. This research makes use of a straightforward CNN architecture that does not incorporate batch normalization or attention layers to enhance the performance of CNN [11,19]. Additionally, simple CNN architecture was utilized by GCNN models to generate feature maps from waveform data [1,23]. The majority of the studies do not incorporate event spatial information, which can boost the accuracy of earthquake detection [13,33]. A few studies have looked into the possibility of event detection using data from multiple stations as compared to data from only one station. In this research, we address the topic of earthquake prediction using a proposed deep learning model that is comprised of two primary components: CNN and GNN. CNN is built with a series of convolutional layers, as well as batch normalization and attention layers. GNN combines the spatial information of the event with the CNN feature map to increase the performance of the model. In this study, events have been collected from multiple stations and station location information is included in both of the analyzed datasets.

## 3. Methods

Graphs have been used to show relationships between chemicals, urban infrastructure, and social networks as nodes and edges. Balsa-Barreiro et al. [38] use spatial networks to map population dynamics nodes representing the human settlements and links showing hierarchies between nodes. Wu et al. proposed a graph convolutional network to detect a social spammer that operates on a directed social graph by explicitly considering three types of neighbors [39]. Due to the lack of spatial ordering in graphs, mathematical operations must be order invariant [24,40]. Graph operations must be able to handle an arbitrary number of nodes and/or edges at any given time since nodes and their connections (edges) may not be fixed. In general, suitable graph operations can be applied to a set of unknown cardinality [34]. Each seismic station represents a node in a graph. Regardless of station location, information is transmitted from the seismic source station to each receiver station. Since local site amplifications may be extremely important in classifying seismic sources, such information should be recorded in each station’s absolute position rather than its relative position [16]. We exclude the links between stations from the study since they are not physically significant, which turns the graph into an unordered collection. When station relationships are relevant, such as in seismic array beamforming, edge information should be provided (which relies on relative locations and arrival times). Our graph comprises nodes with locations and seismic waveform time series. Latitude, longitude, depth, and seismic magnitude are displayed on the graph. GNN predicts graph properties by collecting and analyzing node information [16,27].

In this paper, to enhance the performance of the model to predict earthquakes from multiple stations, we propose an attention-based GCNN structure with batch normalization called BNGCNNAtt. Attention mechanisms have the potential to make feature maps more discriminative, allowing the model to concentrate more on the essential characteristics of the data. In this research, we present an improved attention module that is based on modeling interdependencies between the channels of an extracted feature map [21]. We also present a structure for a CNN in which the approaches of batch normalization and dropout are arranged in a manner that is appropriate for earthquake event prediction. Both batch normalization and dropout regularization reduce overfitting, which could potentially improve the suggested model’s stability and performance. Batch normalization prevents the input distribution from varying between the layers, and dropout regularization prevents overfitting [19].

Figure 1a shows the general architecture of the GCNN model proposed in Ref. [16], where the CNN part takes three channels of input waveform data and extracts a feature map using a convolutional block. This block has convolutional filters to perform a convolutional operation, an activation layer to add non-linearity, and a max-pooling layer to extract useful features from the convolutional operation. The input to the second part of GCNN is the feature map taken from CNN, and the location information is appended with the map. Figure 1b,c shows different variations of the CNN convolutional block. Figure 1b shows a simple layered architecture of convolutional, activation, and max-pooling layers as described above. Figure 1c represents a layered architecture similar to the model architecture proposed by Ref. [20], where the fourth layer is the batch normalization layer. Overfitting is prevented through batch normalization and dropout regularization, which ensure that the input distribution does not differ between layers. Figure 1d also represents a four-layered architecture of the CNN block where the attention layer is added at the end of the layers of Figure 1b. The attention layer makes the feature map more discriminative by concentrating on the most essential characteristics of the data. Figure 1e shows the proposed CNN block that consists of five layers: convolutional, batch normalization, activation, pooling, and attention layers. The proposed architecture is the combination of Figure 1c [16] and Figure 1d [20]. Experiments on two seismic datasets show that our proposed model outperforms the baseline models of GCNN given in Figure 1a–c as it uses both batch normalization and attention layers to achieve high accuracy.

Input to the BNGCNNAtt is a data structure of size $N_{s}\times 510\times 3$ where $N_{s}$ is the total number of stations, which is 50, the sampling rate is 510, and 3 is the number of waveform components. Data is padded with zeros if it was collected by less stations than $N_{s}$. Each waveform begins at its origin time, and the station order is maintained. In order to enhance CNN performance, we have normalized the waveform by its maximum value [35]. The received input is then fed into a stack of five feed-forward convolutional blocks, each of which is made up of three convolutional layers and two-dimensional convolution filters of size $1\times 5\times f_{i}$, where  f={4, 8, 16, 32, 64} is the number of filters in the $i^{th}$ block and ranges from 4 to 64. A linear combination of all the ${\left(k-1\right)}^{th}$ layer components makes up an element of a $k^{th}$ layer feature map. The definition of a convolutional operation is as follows:(1)$$y_{i}^{l+1}\left(j\right)=k_{i}^{l}*M^{l}\left(j\right)+b_{i}^{l},$$
where $y_{i}^{l+1}\left(j\right)$ denotes the input of the $j^{th}$ neuron in the feature map i of layer l+1. $k_{i}^{l}\;$ is the weights of the $i^{th}$ filter kernel in layer l, $M^{j}$ is the $j^{th}$ local region in the layer l and $b_{i}^{l}$ is the bias. An activation function is used to extract the nonlinear features after each convolutional layer. A typical activation function used for this is the Rectified Linear Unit (ReLU), which is defined as:(2)ReLU(x)=max(0,x),
where x represents the convolutional layer’s output. It is a piecewise linear function that returns zero if the input is negative and outputs the value directly if the input is positive.
(3)$$TanH\left(x\right)=\frac{e^{x}-e^{x}}{e^{x}+e^{-x}}=1-\frac{2}{1+e^{2x}}\in \left[-1,1\right].$$

The Euler constant, where e is constant. This activation function has the benefit of being able to return negative values, which is advantageous if the desired output distribution includes negative values.

Before the activation function, batch normalization is added. ReLU introduced the non-linearity, which was followed by a spatial dropout. Three convolutional layers are followed by 1×4 max-pooling with stride 1 and data reduction. Tanh activation function without a dropout is used in the last layer of the final block to preserve the extracted features. The data was reduced by the max-pooling layer from Ns×32×64 to Ns×1×64. It generates a 64-dimensional feature vector for each station Ns. This feature vector is input to an attention layer in order to maintain the essential features while reducing the feature vector. A feature vector Ns×1×66 is produced for each of the Ns stations by adding spatial information (longitude and latitude) to the generated feature vector.

A multi-layer perceptron that recombines spatial information with time-series data makes up the second component of the proposed BNGCNNAtt. It has two convolutional layers, ReLU activation function, and spatial dropout. Applying 1×1 convolutional on Ns feature vector gives an output of size Ns×1×128 . that represents the feature vectors of each graph node (each seismic network station) and encodes spatial information in its elements. Max reduction procedures along the station dimension aggregate node feature vectors into a graph feature vector. For a set $x_{s}=\left\{x_{1},x_{2},x_{3},\ldots ,x_{Ns}\right\}$ the $i^{th}$ element of the reduced vector $\bar{x}$ is represented as follows:(4)$${\bar{x}}_{i}=max\left(\left\{x_{1i},x_{2i},\ldots ,x_{Nsi}\right\}\right).$$

The graph feature vector holds 128 elements with no node-specific information. This vector is used by another MLP of 2 hidden layers with 128 neurons that maps this vector to the output of size 2 (magnitude and depth) after Tanh activation. These predictions are compared with the actual labels scaled between −1 and +1 using root mean squared values. Because all the model components are intimately related to one another by differentiable operations, we train all of the model components together as a single model. All weights in the model are initialized through an orthogonal initialization scheme [41].

## 4. Seismic Dataset

In this study, we use two datasets from Alaska and Japan. The area of study is important because both are the central hub for economic and social activities. Both datasets are made up of three raw waveforms (BHN, BHE, and BHZ) that are directly captured from seismic stations. N was vertically orientated, E was west-to-east, and Z was aligned north-west. For all the available events, we extracted 120 s-long time windows that contained the beginnings of both P- and S-wave arrivals. No consideration was given to earthquakes having a magnitude less than 3.0. Three channels are simultaneously sampled by each station at a resolution of 24 bits. A sac file is created from each waveform file. Waves are filtered at a frequency range from 0.1 to 8.0 Hz and interpolate onto time base 1<t<10 s after the event origin time, over 512 evenly spaced time sample (5 Hz sampling frequency). We consider the waveform’s whole length, from the moment of its beginning to 120 s after its end time.

To gather data, we used the ObsPy Python package. The broadband inventory and seismic catalogues for Japan and Alaska are downloaded by ObsPy. There are 3577 total events in Alaska (AK) dataset, which was collected from 210 stations between 1 January 2020, and 31 December 2021 (2 year). We set the limits for latitude from [52 to 71] and longitude from [−174 to −131] for both seismic stations and event locations, respectively. Figure 2a depicts the locations of these seismic stations as triangles, and Figure 2b shows the event sites as dots.

The map in Figure 2c shows the distribution of stations in the Japan dataset, while the locations of the events are shown in Figure 2d. The network used for data collection is the Japan Meteorological Agency Seismic Network (JMASN). There are a total of 1354 events collected from 19 base stations. This dataset is collected from the period 1st January 2000 to 31 December 2022. The latitude and longitude boundaries are [24° to 44°] and [123° to 143°]. A comparison of both datasets is given in Table 2.

The magnitude distribution of events in a catalog of both datasets is given in Figure 3. In the first row, the Figure 3a histogram shows that the events in the Alaska dataset are not equally distributed by magnitude values. There are a total of 3577 events in the catalog with magnitude values ranging from 3.0 to 8.2. More than 60% of events have a magnitude range of 3.0 to 3.4 while 40% of the events have a 3.5 or greater magnitude. Therefore, it is concluded from the histogram in Figure 3a that the Alaska region frequently faces earthquakes with small magnitude (<3.4). Studies show that machine learning or deep learning models are good to detect small magnitude events. Figure 3b shows the magnitude distribution of events in the Japan dataset. There are a total of 1354 events in the catalog ranging from 3.9 to 7.3 magnitude. More than 80% of events have magnitude values from 4.0 to 4.9 while 20% of events have a larger magnitude than 5.0 or a smaller magnitude than 4.0. From the comparison of both histograms in Figure 3a,b we find that high magnitude events are dominated in the Japan catalog while the Alaska catalog is dominated by events with small magnitudes. We further divided the event catalog into small, medium, and large magnitude events, and their distribution is shown in the histograms in Figure 3c,d for Alaska and Japan. This distribution helps to investigate the event detection with different magnitude values, as large magnitude events are harder to detect than smaller events. The average magnitude value for Alaska events is 3.5 and for Japan is 4.5.

Events with their depth distribution for Alaska as a histogram is shown in Figure 4a and for Japan in Figure 4b. A large portion of the catalog includes events with short depth and small magnitude in Alaska. A total of 2333 events out of 3577 have depths up to 30 km (60% events). In Figure 4b, the distribution of events concerning depth is given for Japan. More than 50% of the events have depths up to 40 km while the catalog is dominant with events with depths of 0 to 10 km. The average value of the depth for Alaska is 33.5 km and for Japan it is 68.0 km.

## 5. Experiment

### 5.1. Training the Network

We created a test set and a training set from the dataset so that we could have two separate and distinct sets. We divided the Alaska and Japan datasets into 80:20 ratios. A total of 20% is for testing the model performance on unseen events, while 80% is for training the model. The training set contains the remaining 2861 (80%) and the testing set contains 716 (20%) events in Alaska. For Japan, the training set has 1083 (80%) events, and the testing set has 271 (20%) events. We used the Adam algorithm for optimization, which kept track of first- and second-order moments of the gradients and was insensitive to diagonal rescaling. We used a learning rate of 10^−4^. A list of experiments has been conducted to analyze the values of the hyperparameters of our model, and the best performing parameter values used in this study are given in Table 3. We ran all of our tests on a system with a NVIDIA GeForce GTX 1080 graphics processor unit, an Intel Core i7-7700 central processing unit, 16 GB of RAM, and the Windows 10 operating system. We use Keras, TensorFlow 2.3, and the CUDA toolkit.

We evaluate the performance of our models using root mean squared error (RMSE) measures. Since the RMSE reflects the degree to which the actual values deviate from the anticipated values, having a number that is as low as possible is preferable. Let the values $y_{1},y_{2},y_{3},\ldots ,y_{n}$ be the ones that were observed in the dataset, and let the values ${\hat{y}}_{1},{\hat{y}}_{2},{\hat{y}}_{3},\ldots ,{\hat{y}}_{n}$ be the ones that were predicted by the model. RMSE can be calculated using the following equation:(5)$$RMSE\left(y,\hat{y}\right)=\sqrt{\frac{1}{N_{samples}}\sum_{i=1}^{N_{samples}}{\left(y-\hat{y}\right)}^{2}}.$$

### 5.2. Result Comparison and Discussion

In the first step of this process, we evaluate the accuracy of our model in relation to those of the baselines. Following the first P wave arrival at any station in the network, we evaluate the models at a set of predetermined times: t = 0.5, 1, 2, 4, 8, and 25 s after the first P arrival. The RMSE values obtained from these evaluations are given in Figure 5. The time is the amount of time that has passed since the first P arrival at any station, and the RMSE is expressed in magnitude units [m.u.]. For the events on the Alaska dataset, initially, RMSE values are high and reduced rapidly for all the models before 5 s, and error is reduced slowly after it. All the models show good performance with a long P wave arrival time. The proposed model outperforms the others because its RMSE value is smaller than the others. On the Japan dataset, the magnitude prediction error of BNGCNNAtt. is again better than the other models. As on the Alaska dataset, initially on a short time of less than 5 s, all the models have error values between six and five but after it, all the models improved their prediction accuracy. Overall, the RMSE values on both datasets show that the proposed model significantly performs better than the baseline models for magnitude estimation. In many applications, the accuracy of magnitude estimation methods for large events is of utmost importance. This is because such events are more likely to cause catastrophic damage.

We compare the RMSE of BNGCNNAtt. to other models, in the groups of small, medium, and large events according to the catalog magnitude. The distribution of events of both datasets into small, medium, and large groups is given in Figure 3c,d. Table 4 shows the RMSE values with the arrival time of the P wave for the Alaska dataset. Values in Table 4 show that all the models with low RMSE values estimate magnitude values significantly more on small magnitude events than on medium and large magnitude events. In addition, these models show good performance over a large time value instead of a small time value. As the time increases, the RMSE values decrease. The simple GCNN model shows the least performance even with batch normalization. The attention mechanism shows more significant performance than normal GCNN and GCNN with batch normalization. Comparatively, the proposed model BNGCNNAtt. outperforms the other three models. Our proposed model shows better performance than others on small, medium, and large magnitude event estimation. Our model takes advantage of both batch normalization and attention mechanisms with quick and reliable magnitude value estimation. The performance of all the models to estimate large magnitude events is not like that for small magnitude event estimation. It is because in our Alaska dataset, the number of large events is small, which may lead to insufficient training of the models.

For the Japan dataset, the estimated performance of all the models is almost the same for large and medium-magnitude events as the results given in Table 5. Our model takes advantage of batch normalization and attention mechanism techniques to predict medium and large magnitude events very well. On small magnitude events, the performance of all the models is significantly lower than large and medium magnitude events that are different than Alaska. GCNN, BNGCNN, and GCNNAtt. estimate the magnitude value similarly. It is because of the dense distribution of large and medium magnitude events and the sparse distribution of small magnitude events. Events with large magnitude values are better predicted than medium and large magnitude events. Overall, BNGCNNAtt. significantly outperforms the others on both the Japan and Alaska datasets to predict small as well as large magnitude events.

We applied our model to both datasets to predict the depth of events of small, medium, or high magnitude. Figure 6 shows the estimation of depth in kilometers for Alaska in Figure 6a and Japan in Figure 6b. All the models agree that the depth of high magnitude events is harder to detect than smaller magnitude events. The RMSE values show that the proposed model detects the depth efficiently for both small and large magnitude events. As the average depth of events in the regions of Alaska is very low when compared to Japan events (35 km approximately), all the models show superior accuracy in Alaska than in Japan for small and medium magnitude events. However, for the Japan region, where medium and high magnitude events dominate over smaller events, all the models do not have high performance and produce approximately the same results. Even batch normalization and attention mechanisms do not help to improve performance. A significant improvement in depth estimation is noted for the proposed model when it is applied to all groups of magnitude events.

## 6. Conclusions

In this study, we proposed a deep learning-based model batch normalized graph convolutional neural network with an attention mechanism to estimate the magnitude and depth of an earthquake event. For the experiments, we used two seismic datasets for Alaska and Japan. Alaska is dominated by events with small magnitude and small depth while Japan is dominated by high magnitude and high depth events. We preprocessed the dataset and categorized the events into three groups—small, medium, and high—according to their magnitude. Following the first P wave arrival at any station in the network, we evaluate the models at a set of predetermined times: t = 0.5, 1, 2, 4, 8, and 25 s after the first P arrival. Our results show that the proposed model efficiently detects the event magnitude (either small or high) and depth using attention and batch normalized techniques. Our model outperformed the other baseline models by achieving significant scores of RMSE. It achieved 1.62, 2.54, and 2.87 RMSE scores for Alaska events and 2.21, 2.43, and 2.66 for Japan events for small, medium, and large magnitude events, respectively. The proposed model provides accurate estimates of the depth and magnitude of small, medium, and large events, as evidenced by low RMSE values, greatly outperforming the three baseline models on both datasets.

## Figures and Tables

**Figure 1 sensors-22-06482-f001:**
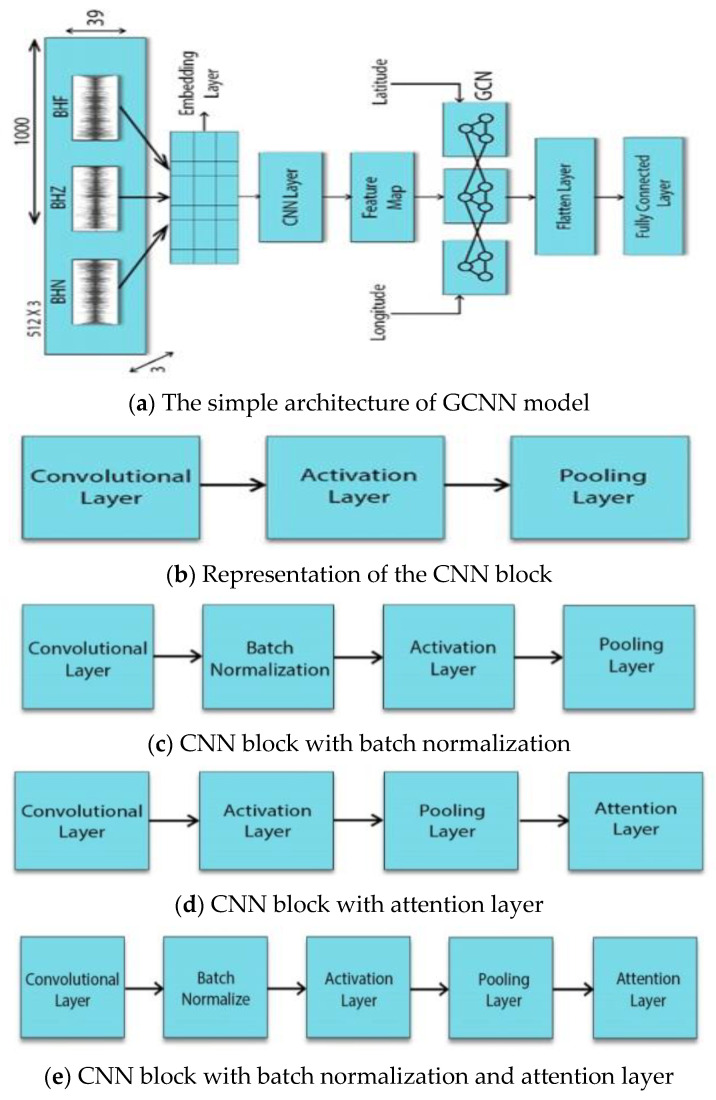
The proposed architecture of the seismic graph convolutional neural network (SGCNN).

**Figure 2 sensors-22-06482-f002:**
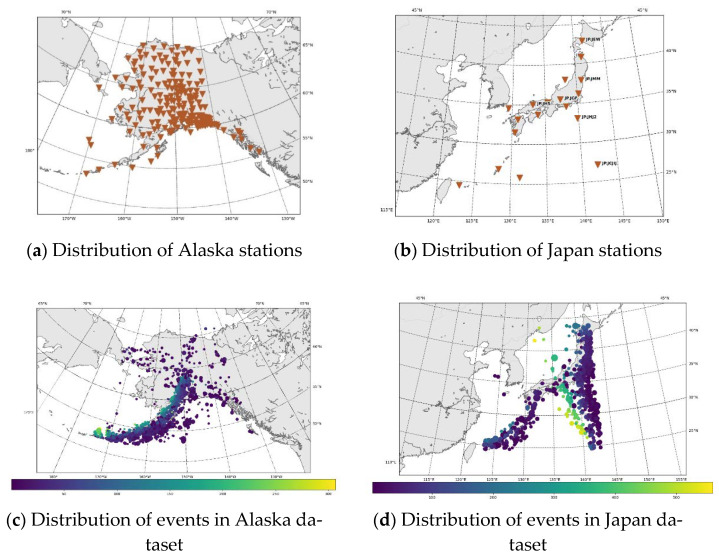
The study’s spatial distribution information for the stations and events that were examined.

**Figure 3 sensors-22-06482-f003:**
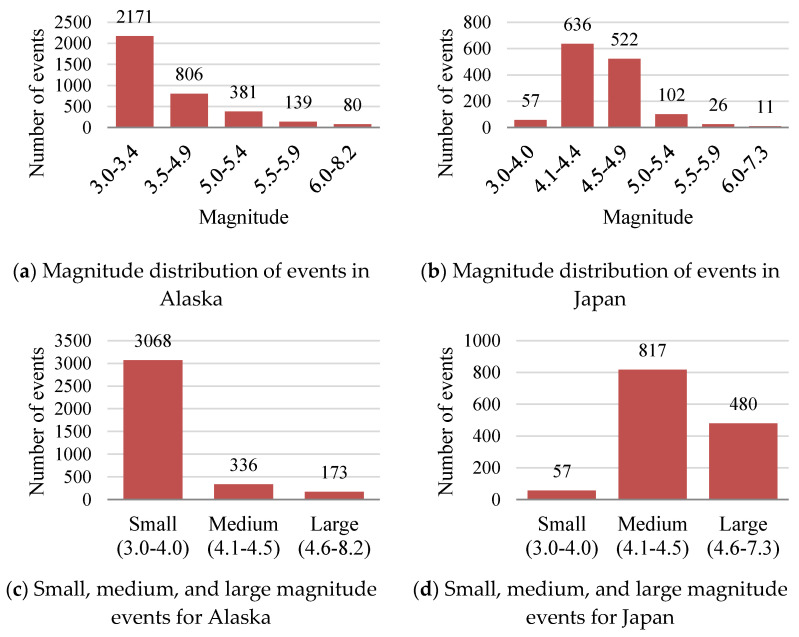
Magnitude distribution of events for both Alaska (**left column**) and Japan (**right column**). The top row shows the overall magnitude distribution of events while the bottom row shows magnitude distribution in three categories: small, medium, and large magnitudes.

**Figure 4 sensors-22-06482-f004:**
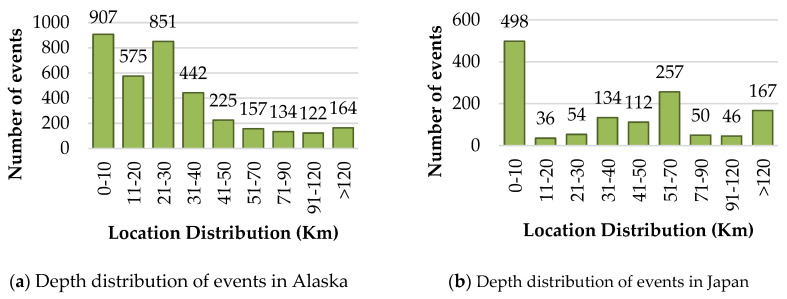
Histograms showing the depth distribution of the Alaska and Japan datasets.

**Figure 5 sensors-22-06482-f005:**
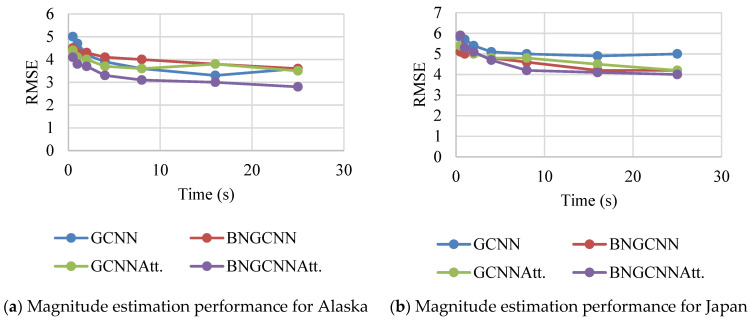
Magnitude prediction performance of the BNGCNNAtt. with other three baseline models after the first P arrival at the station.

**Figure 6 sensors-22-06482-f006:**
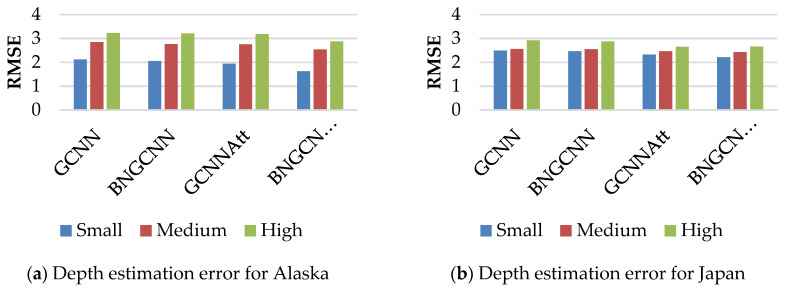
RMSE values achieved by the proposed model and others to predict the depth of small, medium, and large magnitude events for both the Alaska and Japan datasets.

**Table 1 sensors-22-06482-t001:** Summary of the studies that investigated earthquake prediction using different types of deep learning models. In the fourth column, BN stands for batch normalization and Att. stands for attention mechanism.

Study	Used	Spatial Info	BN/Att.	Data	Station
[21]	CNN	No	BN/Att.	South Korea	Single
[29]	SVMR	No	No	Bogota, Colombia	Single
[14]	CNN	No	No	IRIS	Single
[27]	CNN and Graph	No	No	MeSO-Net Japan	Multiple, single
[28]	CNN and Transformer	Yes	Att.	Japan, Italy	Multiple
[4]	GCNN		No	--	Multiple
[16]	GCNN	Yes	No	California	Multiple
[15]	CNN + LSTM + BiLSTM + Transformer	No	Att.	STEAD	Single
[24]	GCNN	Yes	No	Italy and California	Multiple
[23]	Deep CNN		No	CARABOBO	Single
[35]	CNN	No	No	Central Italy	Multiple

**Table 2 sensors-22-06482-t002:** Statistics of both the Alaska and Japan datasets.

Dataset Detail	Alaska (AK)	Japan (JP)
Period	2020–2021	2000–2022
Min. and Max. Latitude	[52° to 71°]	[24° to 44°]
Min and Max. Longitude	[−174° to −131°]	[123° to 143°]
Minimum magnitude	3.0	3.0
Number of events	3577	1354
Number of stations	210	19
Filter the waveform	0.1–8 Hz	0.1–8 Hz
Time-base	1<t<101	1<t<101
Even spaced time sample	512 Hz	512 Hz
Scaled Min. max. source depth	0 to 30 km	0 to 30 km
Scaled magnitude	3–6	3–6
Data split	80–20	80–20

**Table 3 sensors-22-06482-t003:** Detail of the hyperparameter values of the BNGCNNAtt. model.

Dataset Detail	Alaska (AK)	Japan (JP)
Batch Size	32	32
Number of Epochs	500	300
Dropout	0.4	0.6
Training:validation:testing	60:20:20	60:20:20

**Table 4 sensors-22-06482-t004:** RMSE scores were obtained from the small, medium, and large magnitude events prediction from the Alaska dataset after P-wave arrival time.

Time	Small				Medium				Large			
	GCNN	BNGCNN	GCNNAtt.	BNGCNNAtt.	GCNN	BNGCNN	GCNNAtt.	BNGCNNAtt.	GCNN	BNGCNN	GCNNAtt.	BNGCNNAtt.
0.5	2.42	2.62	2.33	2.82	3.12	3.82	3.42	3.18	5.48	5.38	5.32	5.45
1	2.35	2.44	2.38	2.75	3.16	3.63	3.22	2.98	5.42	5.35	5.26	5.38
2	2.28	2.31	2.42	2.71	2.91	3.54	3.15	2.76	5.37	5.31	5.29	5.26
4	2.25	2.39	2.23	2.63	2.73	3.21	3.08	2.59	5.36	5.28	5.25	5.18
8	2.17	2.11	2.12	2.48	2.65	3.04	2.94	2.42	5.32	5.22	5.22	5.11
16	2.27	2.03	2.04	2.35	2.36	2.79	2.85	2.28	5.25	5.12	5.13	5.02
25	2.2	2.21	2.11	1.94	2.32	2.63	2.71	2.04	5.21	5.04	5.03	4.93

**Table 5 sensors-22-06482-t005:** RMSE scores were obtained from the small, medium, and large magnitude events prediction from the Japan dataset after p wave arrival time.

Time	Small				Medium				Large			
	GCNN	BNGCNN	GCNNAtt.	BNGCNNAtt.	GCNN	BNGCNN	GCNNAtt.	BNGCNNAtt.	GCNN	BNGCNN	GCNNAtt.	BNGCNNAtt.
0.5	3.64	3.43	3.41	3.38	2.86	2.44	2.41	2.38	2.84	2.62	2.63	2.54
1	3.62	3.35	3.36	3.32	2.68	2.36	2.38	2.31	2.63	2.59	2.58	2.37
2	3.57	3.32	3.25	3.27	2.57	2.54	2.52	2.32	2.59	2.53	2.54	2.32
4	3.45	3.28	3.29	3.22	2.51	2.48	2.46	2.28	2.54	2.52	2.42	2.38
8	3.41	3.27	3.27	3.16	2.48	2.40	2.43	2.24	2.42	2.44	2.44	2.22
16	3.36	3.22	3.21	3.08	2.42	2.38	2.40	2.29	2.38	2.32	2.36	2.21
25	3.32	3.21	3.23	2.93	2.38	2.31	2.28	2.18	2.35	2.28	2.31	2.09

## Data Availability

Not applicable.

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
