# Peer review of "An Early Warning System for Earthquake Prediction from Seismic Data Using Batch Normalized Graph Convolutional Neural Network with Attention Mechanism (BNGCNNATT)"

_sensors, 2022, doi:10.3390/s22176482_

Round 1

Reviewer 1 Report

The review of research paper “An Early Warning System for Earthquake Prediction from Seismic data using Batch Normalized Graph Convolutional Neural Network with Attention Mechanism (BNGCNNAtt.)” authored by Muhammad Atif Bilal, Yanju Ji, Yongzhi Wang, Muhamamd Pervez Akhter, and Muhammad Yaqub.

As I understand, this manuscript proposes a novel deep learning model (BNGCNNAtt.) This model can predict the depth and magnitude of an earthquake event at certain number of seismic stations in certain number of locations. The main process of model prediction includes: preprocessing the raw seismic waveform data, extracting the feature map by using CNN, and predicting the seismic event based on the features and event location information extracted from GNN. The model proposed in this manuscript outperforms the three baseline deep learning models on both real case datasets to provide a relatively accurate estimation of the depth and magnitude of seismic events.

My comments and suggestions are as following:

1. The progress of many technologies or methods is developed step by step. Even if there are some problems in the previous research, these foundations are also an important reference for subsequent technological progress. The authors mentioned in the introduction that deep learning algorithms have been used in the past to predict earthquakes, but they have significant drawbacks. This expression is inappropriate.

2. When the abbreviation first appears in the abstract, the full name should be given (CNN, GNN).

3. I strongly suggest that the authors condense the contents of section 2 and put them into Section 1 Introduction and Section 3 Methods.

4. L320: Why the authors do not used activation function in the last convolutional layer?

5. L321: The authors used Tanh activation function without a dropout in the last layer of the final block to preserve the extracted features. Have the authors tested other activation functions? Is the effect of the Tanh activation function without a dropout optimal?

6. L345: Why the authors set the weights in the model to their default values. Is this reasonable? What is the default value? The author should give the necessary explanation.

7. L443: What is the basis for setting of predetermined times? 

8. L443: Table.4 and Table.5 show that the RMSE scores obtained from BNGCNNAtt. are not significantly better than that of other models, which is basically two digits after the decimal point. The author should give the necessary explanation. Additionally, what will happen if some more predetermined times are added.

9. The author needs to pay attention to the accuracy of some words, such as accurately predict earthquakes. Because up to now, the accurate prediction of earthquakes is still an unsolved worldwide problem.

10. Is the template of the manuscript sensor or Applied Sciences?

Author Response

We are thankful to all the reviewers and editors for their valuable feedback that help and guide us to improve the quality of our manuscript. We have tried our best to fix all the identified gaps, and technical issues raised by the honorable reviewers. To understand the changes in the manuscript, we have highlighted the text in the manuscript.

Reviewer 2 Report

An Early Warning System for Earthquake Prediction from Seismic data using Batch Normalized Graph Convolutional Neural Network with Attention Mechanism (BNGCNNAtt.)

In this manuscript, the authors propose a novel deep learning model, batch normalized graph convolutional neural network with attention mechanism (BNGCNNAtt.), that can successfully predict the depth and magnitude of an earthquake event at any number of seismic stations in any number of locations. After preprocessing the raw seismic waveform data collected from multiple seismic stations, CNN is applied to extract the feature map. The proposed model is tested with data from Japan and Alaska, which are seismically complimentary regions (no. of events, depth, and magnitude distribution). The proposed model significantly outperforms the three baseline deep learning models on both datasets to provide an accurate estimation of the depth and magnitude of small, medium, and large-magnitude events. It achieves 2.8 and 4.0 RMSE values in Alaska and Japan for magnitude prediction, and 2.87 and 2.66 RMSE values for depth prediction.

I have read the manuscript and the topic is interesting for a specific audience. The structure is quite optimal. However, I have detected some aspects that should be improved according to my following suggestions:

§  First point, the abstract is quite long and unclear. This is not well focused on the method and objectives. The authors start with “In this manuscript, the authors propose a novel deep learning model, batch normalized graph convolutional neural network with attention mechanism (BNGCNNAtt.), that can successfully predict the depth and magnitude of an earthquake event at any number of seismic stations in any number of locations”. I would suggest to rephrase totally this: “The authors implement a deep learning model for predicting predict earthquakes. This method is based on a mechanism….”. For me, it is particularly confusing the presentation of the method: “After preprocessing the raw seismic waveform data collected from multiple seismic stations, CNN is applied to extract the feature map. The attention mechanism captures long-range feature interactions used by the model to focus on important features. Batch normalization improves model training by applying normalization to every mini-batch to reduce training epochs. GNN with extracted features and event location information predicts the event information accurately”, but also how the results are presented: “It achieves 2.8 and 4.0 RMSE values in Alaska and Japan for 32 magnitude prediction, and 2.87 and 2.66 RMSE values for depth prediction”. What do you mean by these results? How much are improving these results in terms of qualitative in comparison to other traditional methods?

§  Particularly, I think the use of the acronym BNGCNNAtt for the method here implemented “batch normalized graph convolutional neural network with attention mechanism” is too long, unclear, unpractical and confusing. Why do you use an acronym so long?

§  This sentence is unclear and reiterative (using twice the word features): “The attention mechanism captures long-range feature interactions used by the model to focus on important features”. The same with the next one: “Batch normalization improves model training by applying normalization to every mini-batch to reduce training epochs.” You should not use the same word for defining and problem and proposing a solution.

§  Also in the abstract and in the introduction, the authors argue “Japan and Alaska, which are seismically complimentary (no. of events, depth, and magnitude distribution)”. After that, you argue “two high-risk seismic datasets, Japan and Alaska”. What do you mean with “seismically complementary” regions? (Probably what I read in the manuscript after: “Alaska is dominant by events with small magnitude and small depth while Japan is dominant by high magnitude and high depth events.”) I do not agree that you can refer to “seismically complementary” because of that. You could say that they have different seismic dynamics, for example. Please extend this argument and be consistent for why you select these particular regions. Also the part in parenthesis should be part of the explanation. I also recommend not to use acronyms like “no.” instead of number.

§  About the results, the authors argue, “It achieves 2.8 and 4.0 RMSE values in Alaska and Japan for magnitude prediction, and 2.87 and 2.66 RMSE values for depth prediction.” Really, I cannot understand well this point. What do you mean with these values/estimator in simple words and generally understandable? I know of course the RMSE factor, but I do not get the point here.

§  The authors start in the introduction with “One of the goals of earthquake hazard analysis is to limit the impact of earthquakes on society by minimizing damage to buildings and infrastructure. Predicting seismic events from continuous data and applying it in real-time to early warning systems or analyzing it offline to look for previously unpredicted earthquakes is an important task”. I would recommend you to add a first paragraph more extensive for a broad audience referring to natural hazards and catastrophe mapping systems. At the end, the earthquakes are just one more, but we could refer to more. There is extensive literature in geography and geospatial sciences about these events and also about the techniques used. For example, you can refer to LiDAR (laser scanning), Doppler, or remote sensing techniques. Let me suggest some particular studies such J. Balsa-Barreiro (2019). LiDAR for management in natural disasters and catastrophes. In: K.G. Greene (ed.): Government Briefing Book: Emerging Technology & Human Rights, 1(Aug): 11, and other studies focused on LiDAR techniques such “Airborne light detection and ranging (LiDAR) point density analysis.”

§  The authors argue: “Several recent studies show that deep learning models on raw seismic waveforms can effectively predict the earthquake parameters like magnitude [11–13], location [13,14], and peak ground acceleration [11].” Before this point you should introduce why deep learning must be applied in EEW systems to earthquake detection and what particular advantages or gaps are covered.

§  In Line 72, the authors argue “Batch normalization is a technique that is used to make the training of deep learning models more consistent and faster [4].” After that, in the introduction, the authors argue “Batch normalization is a technique that can be used to speed up and improve the accuracy of deep learning model training [15,16]. The goal of the batch normalization method during training is to normalize the layer output by utilizing the statistics of each mini-batch size”. What do you mean with “to make the training”? But batch methods also allow to reduce the computational costs by optimizing the resources, right?

§  In line 74, the authors refer to CNN -> “In CNN, a feature map is obtained from a convolutional layer that has contextual information among the features but does not focus on the important feature”, but they did not introduce before what CNN means. They do some lines ahead, in line 81.

§  In line 81, the authors argue “On recent years, convolutional neural networks (CNN) [5,13], graph neural networks (GNN) [19], and their ensemble model graph convolutional neural networks. (GCNN) [1,5] have shown promising results for earthquake prediction. To analyze data from networks, graph neural networks (GNN) have been developed specifically [20].” Why before the acronym GCNN there is a “.”?  Again, this sentence is reiterative and unclear “To analyze data from networks, graph neural networks (GNN) have been developed specifically

§  In line 146, the authors argue: “The effectiveness of models like decision trees, support vector  machines, and naive Bayes is significantly impacted by these feature selection techniques [14,26].” Do you mean “Bayesian Methods” with “Bayes”? What do you mean with “these selection techniques”? It is unclear in my eyes.

§  “On a range of tasks, including image processing [4,5], text classification [2,3], speech recognition [6,7], and others, deep learning models have shown performance that is superior to that of traditional machine learning models.” The argument “superior” needs to be more explicit in a scientific context. Superior in accuracy, processing times, computations, etc?

§  In the line 166, the authors argue “[29] used CNN…”. I strongly recommend to avoid this way of citation. You should use the name such “Jozinović et al. [29] used CNN”.

§  In the line 245, the authors refer to “Graphs have been used to show relationships between chemicals, urban infrastructure, and social networks as nodes and edges. Due to the lack of spatial ordering in graphs, mathematical operations must be order invariant [20,33]. Graph operations must be able to handle an arbitrary number of nodes and/or edges at any given time since nodes and their connections (edges) may not be fixed. In general, suitable graph operations can be applied to a set of unknown cardinality [28].” After the first sentence, I recommend you to refer to some relevant studies in the use of graphs, which is extensive in complex systems and network science. This argument (“Due to the lack of spatial ordering in graphs”) is a little bit imprecise because the so-called spatial networks have into account the geographical location of nodes. You should refer to some relevant study by using spatial networks such were used by Morales and Lois for Mapping Population Dynamics at Local Scales (demography). Please include this and some additional studies to this point.

§  In the conclusions, the authors argue “We preprocessed the dataset and then divided the events into three groups of small, medium, and high magnitude events”. The authors should rephrase this by using a more scientific vocabulary such “we categorize the events in three groups according to their magnitude”

§  Again in the conclusions, the authors should rephrase the text and be more clear with the output.

§  What the authors mean with “BN/Att.” in Table 1. Please, explain this with more detail.

§  I miss the units represented in both axis in the charts 3, 4 and 5. You should include them.

Author Response

(The authors gave the same response as above.)
